# Diversity of oligomerization in *Drosophila* semaphorins suggests a mechanism of functional fine-tuning

Daniel Rozbesky [1], Ross A. Robinson[1,2], Vitul Jain [1], Max Renner[1], Tomas Malinauskas [1], Karl Harlos [1], Christian Siebold [1] & E. Yvonne Jones [1]

Semaphorin ligands and their plexin receptors are one of the major cell guidance factors that trigger localised changes in the cytoskeleton. Binding of semaphorin homodimer to plexin brings two plexins in close proximity which is a prerequisite for plexin signalling. This model appears to be too simplistic to explain the complexity and functional versatility of these molecules. Here, we determine crystal structures for all members of *Drosophila* class 1 and 2 semaphorins. Unlike previously reported semaphorin structures, Sema1a, Sema2a and Sema2b show stabilisation of sema domain dimer formation via a disulfide bond. Unexpectedly, our structural and biophysical data show Sema1b is a monomer suggesting that semaphorin function may not be restricted to dimers. We demonstrate that semaphorins can form heterodimers with members of the same semaphorin class. This heterodimerization provides a potential mechanism for cross-talk between different plexins and co-receptors to allow fine-tuning of cell signalling.

---

[1] Division of Structural Biology, Wellcome Centre for Human Genetics, University of Oxford, Roosevelt Drive, Oxford OX3 7BN, UK. [2] Present address: Immunocore Ltd, Milton Park, Abingdon OX14 4RY, UK. Correspondence and requests for materials should be addressed to D.R. (email: daniel@strubi.ox.ac.uk) or to E.Y.J. (email: yvonne@strubi.ox.ac.uk)

The semaphorins are the largest family of evolutionarily conserved axon guidance cues. A growing body of evidence shows that beyond axon guidance, semaphorins play essential roles in many tissues for myriad activities, including angiogenesis, cardiovascular development, cell migration, tumour progression, immune responses and bone homeostasis (reviewed in ref. [1]). The semaphorins signal through their main receptors, plexins, which are single-spanning transmembrane proteins expressed in a variety of cell types, including neurons, cancer cells or endothelial cells. Sequence analyses indicate that the semaphorins and plexins are characterized by an N-terminal region designated the sema domain. This feature is also common to the MET receptor, a receptor tyrosine kinase triggered by hepatocyte growth factor/scatter factor[2]. The first crystal structure of semaphorins revealed that the sema domain forms a dimer[3,4]. In the crystal structures, the sema domain is followed by a cysteine-rich knot (the PSI domain) and in class 3, 4 and 7 structures, an Ig-like β-sandwich domain[5]. A bivalent 2:2 architecture, common to phylogenetically distinct semaphorin–plexin complexes, highlighted that semaphorin-mediated dimerization of plexin receptors is a central mechanism for triggering signal transduction[6–8]. The ligand–receptor interaction is mediated through their respective N-terminal sema domains. Subsequent structural and functional studies showed that the ten-domain plexin ectodomain can exist in a ring-like conformation[9,10]. The ring-like conformation is implicated in plexin autoinhibition and also suggests an exquisite mechanism for the interplay between the ectodomain and intracellular region in the activation of signal transduction[9].

More than 29 semaphorins have been identified to date, and they can be classified into eight classes based on sequence similarity and domain organization[11]. Classes 1 and 2 are invertebrate semaphorins, class V are viral-encoded semaphorins and classes 3 through 7 are found in vertebrates. Class 5 is the only one that includes both vertebrate and invertebrate semaphorins. Semaphorins exist as secreted (classes 2, 3 and V), single-membrane spanning (classes 1, 4, 5 and 6) or GPI-anchored proteins (class 7). While the vertebrate semaphorin family includes a large number of semaphorins, only five semaphorins (Sema1a, Sema1b, Sema2a, Sema2b and Sema5c) have been identified in *Drosophila*. Also, there are only two plexins (PlexA and PlexB) in *Drosophila*. Class 1 semaphorins (Sema1a and Sema1b) are transmembrane proteins and bind PlexA, while class 2 semaphorins (Sema2a and Sema2b) are secreted proteins and bind PlexB[12–14]. Sema5c has been shown to bind PlexA and intriguingly, the Sema5c–PlexA interaction has been reported to be crucial for collective migration of follicular cells in a contact-dependent manner[15]. Unlike class 1 and 2 semaphorins, the Sema5c ectodomain is distinctive in containing a series of seven additional thrombospondin type 1 domains.

Sema1a has been reported initially as a repulsive guidance cue mediating the defasciculation of motor axon bundles in the fly embryo[16]. Similarly, Sema1b has been found to be equally capable of repelling motor axons when expressed ectopically[12]. The genetic analysis provided compelling evidence that PlexA is a neuronal receptor for both Sema1a and Sema1b[12]. A transmembrane protein off-track has been shown as a component of the Sema1a–PlexA signalling in mediating the defasciculation of embryonic motoneuron axons[17] and lamina-specific targeting of axons in the visual system[18]. Sema1a–PlexA repulsive signalling can be further modulated by a secreted heparan sulfate proteoglycan perlecan, an extracellular matrix component[19]. Intriguingly, Sema1a has also been found to mediate reverse signalling in a wide range of cellular responses, including synapse formation, dendritic targeting and axon–axon repulsion as well as attraction[20–23].

Both Sema2a and Sema2b signal through the same receptor, PlexB, during embryonic CNS development; however,

remarkably, they have been reported to serve opposing guidance functions (repulsion vs. attraction)[14]. In the developing *Drosophila* olfactory circuit, both Sema2a and Sema2b are expressed in a gradient that specifies the olfactory receptor neuron axons trajectory choice through PlexB[24]. Intriguingly, Sema1a is distributed in a gradient opposing the Sema2a and Sema2b gradients within the antennal lobe, and both Sema2a and Sema2b have been reported to signal through Sema1a suggesting cooperation between secreted and transmembrane semaphorins[25]. Previous work emphasized the importance of the spatiotemporal differential distribution of endogenous PlexB, suggesting that different PlexB levels in the developing antennal lobe induce divergent behaviours of growth cones in response to external semaphorin cues[26].

Although *Drosophila* semaphorins are a functionally particularly well-characterized group of semaphorins, no mechanistic view and structural data have been reported for these molecules to date. Here, we present crystal structures for all members of *Drosophila* class 1 and 2 semaphorins. A structural comparison provides insight into molecular differences and similarities among the fly semaphorins concerning their function, evolution and binding specificity to their plexin receptor. We describe distinctive structural features that allow us to propose mechanisms that may explain functional versatility and complexity of these molecules.

## Results

**Crystal structures of class 1 and class 2 semaphorins.** We determined crystal structures of *Drosophila* Sema1a$_{1-2}$, Sema1b$_{1-2}$, Sema2a$_{1-3}$ and Sema2b$_{1-3}$ to a 3.6-, 2.8-, 2.1- and 2.5 -Å resolution, respectively (Fig. 1a–d, Supplementary Table 1 and Supplementary Fig. 1). All structures contain the sema domain composed of a seven-bladed β-propeller fold, which is followed by a PSI domain. The secreted semaphorins Sema2a and Sema2b also contain an Ig-like domain positioned C-terminally to their PSI domains. In the crystal, Sema1a$_{1-2}$, Sema2a$_{1-3}$ and Sema2b$_{1-3}$ form a disulfide-linked dimer, while Sema1b$_{1-2}$ is a monomer and the lattice packing provides no evidence for Sema1b$_{1-2}$ dimerization within the crystal. More generally, analyses of the packing for our *Drosophila* semaphorin crystal structures do not highlight any recurrent interfaces or interaction modes of potential biological significance.

The overall architecture of *Drosophila* semaphorins is similar to all previously determined vertebrate semaphorins. In accordance with predictions from sequence and the observed protein molecular weights, the crystal structures of *Drosophila* semaphorins show a high level of N-linked glycosylation. For example, the Sema2a construct used for crystallization contains eight potential sites for N-linked glycosylation (N95, N163, N190, N229, N314, N401, N563 and N658) and the N-linked glycans were clearly visible and unambiguously fitted into the electron density at N95, N163, N190, N229 (only chain B of the dimer), N314 and N563. Of these sites, N314 is highly conserved across classes and species. In one of the following sections, we describe the role of this glycan in semaphorin dimerization.

A structure-based phylogenetic tree indicates that class 2 semaphorins cluster separately from the vertebrate semaphorins and the class 1 semaphorins are most similar to mouse Sema6A (Fig. 1e), consistent with a previously reported sequence-based phylogenetic tree that relates class 1 and class 6 semaphorins[27]. The close evolutionary relationship between *Drosophila* class 1 semaphorins and vertebrate class 6 semaphorins is also supported by their identical ectodomain organization, binding specificity to the same class of plexin receptors and also by their shared capacity to mediate reverse signalling (reviewed in ref. [28]).

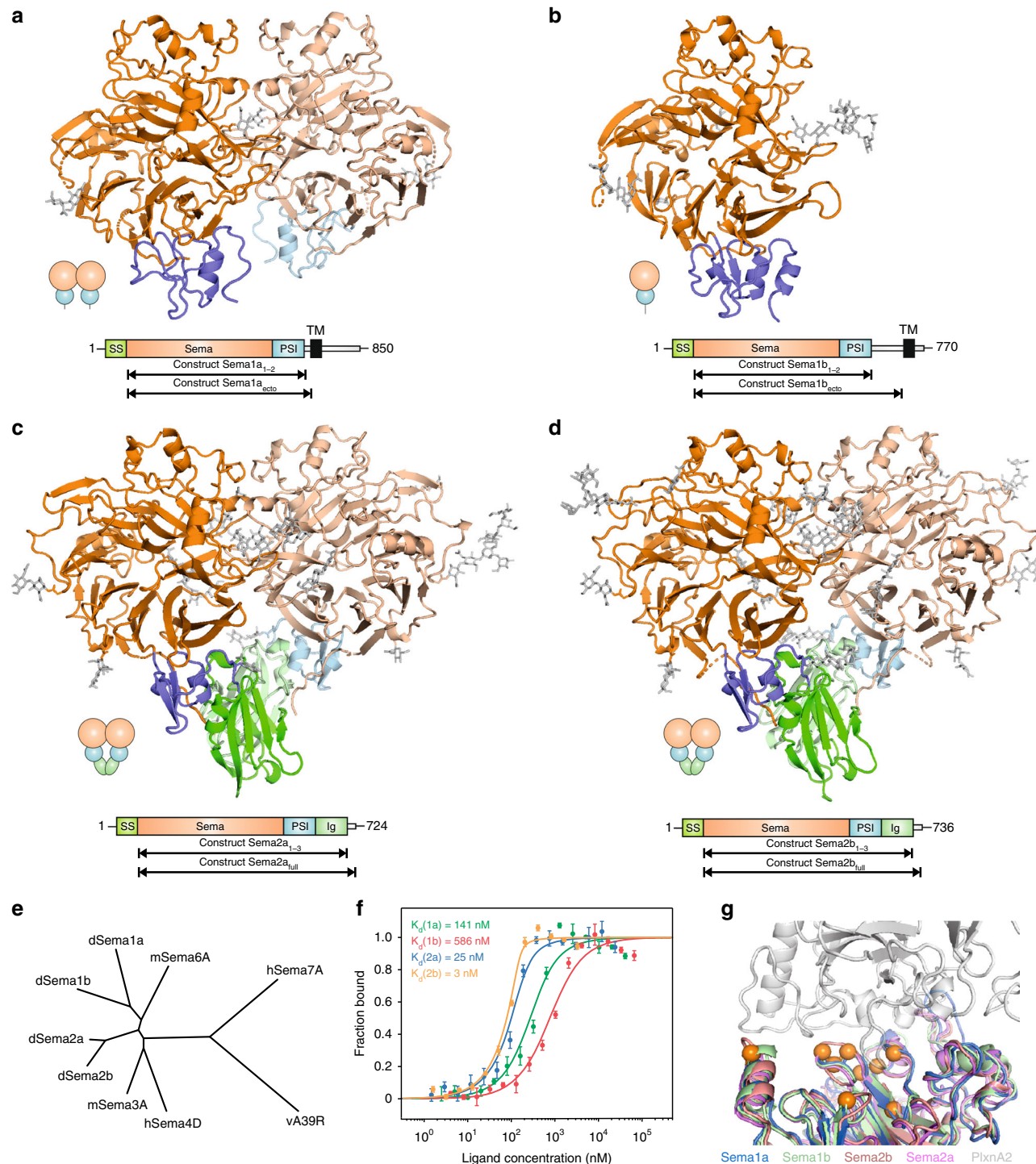

**Fig. 1** Crystal structures of class 1 and class 2 semaphorins. **a**–**d** Ribbon representation of Sema1a$_{1-2}$ (**a**), Sema1b$_{1-2}$ (**b**), Sema2a$_{1-3}$ (**c**) and Sema2b$_{1-3}$ (**d**). N-glycans are shown in stick representation (grey). Chains A and B are coloured in dark and light shades, respectively. Schematic domain organizations of semaphorins are shown below the ribbon representation. SS, signal sequence; TM, transmembrane region. **e** Structure-based phylogenetic tree of the sema domains constructed from all available structures of sema domains (mSema3A pdb code 1q47, hSema4D pdb code 1olz, mSema6A pdb code 3okw, hSema7A pdb code 3nvq, vA39R pdb code 3nvn, Sema1a, Sema1b, Sema2a and Sema2b described here). The phylogenic tree was calculated with SHP[55]. **f** Microscale thermophoresis binding experiment for PlexA$_{1-4}$-mVenus and Sema1a$_{ecto}$-Fc (green) or Sema1b$_{ecto}$ (red), and PlexB$_{1-4}$-mVenus and Sema2a$_{full}$ (blue) or Sema2b$_{full}$ (orange). Error bars represent s.d. of three technical replicates. Source data are provided as a Source Data file. **g** The putative binding site for plexin was identified by superposition of *Drosophila* semaphorins with mouse Sema6A in complex with PlxnA2 (white). The architecture of the putative binding site is very similar among *Drosophila* semaphorins. Class-dependent binding specificity to plexin receptors is probably determined by nine residue substitutions (orange) that are spatially spread throughout the putative interface. The structures were superposed via their sema domains

Next, we probed the flexibility of the fly semaphorins using explicit-solvent, classical molecular dynamics simulations. We observed generally high conformational stability, with root mean square fluctuations (RMSFs) of below 0.15 nm for the majority of residues over the course of the simulations (Supplementary Fig. 2a, b). Slightly increased inherent flexibility was observed for solvent-exposed loops extending from the bottom and the top face of the β-propeller. This observation prompted us to explore whether higher flexibility of these loops might contribute an entropic penalty that prevents Sema1b dimerization. However, there was no significant difference in dynamic behaviour between monomeric Sema1b$_{1-2}$ and dimeric Sema2a$_{1-3}$, indicating that dimerization does not increase structural stability. The extended β2C–β2D and β6C–β6D loops in Sema1b$_{1-2}$ appeared to have a higher degree of flexibility relative to the rest of the protein. The simulations revealed a low inter-domain flexibility that is also obvious from the structural superposition of two independent protein chains observed in Sema2a crystal (Supplementary Fig. 3a). The Ig-like domain can reorientate relative to the sema–PSI domains by 14° about a hinge point in the PSI-Ig linker.

**Unique interclass variations in *Drosophila* semaphorins.** Although the overall architecture of the sema domain is conserved among the *Drosophila* semaphorins, there are substantial interclass variations, especially the bottom face of the sema domain. In class 2 semaphorins, the β3A–β3B and β3C–β3D loops of blade 3 protrude prominently from the bottom face of the β-propeller (Supplementary Fig. 3b). The Sema2 β3A–β3B loop is longer by seven residues than that of Sema1s and has a distinctive helical arrangement (α2A, Supplementary Fig. 1). Sequence analysis shows that this extended β3A–β3B loop is unique for insect class 2 semaphorins and *Caenorhabditis elegans* Sema2a (Supplementary Fig. 4). The β3C–β3D loop forms a distinctively extended hairpin displaying several prominent polar residues; sequence analysis reveals that this loop is unique for insect class 2 semaphorins and also *C.elegans* SMP-1, SMP-2 and Sema2a. Notably, both the β3A–β3B and β3C–β3D loops align with the major binding interface between the sema domain of the MET receptor and its ligand the hepatocyte growth factor β-chain[29] (Supplementary Fig. 3c). In class 1 semaphorins, two distinct loops with no obvious structural and functional role are substantially extended from the bottom face of the β-propeller, β5A–β5B and β7A–β7B in Sema1a, and β2C–β2D and β6C–β6D Sema1b (Supplementary Fig. 3b). The latter loops in Sema1b also showed a higher level of fluctuations relative to the rest of protein in the molecular dynamics simulations (Supplementary Fig. 2b).

*Drosophila* semaphorins also show interclass variations in the distribution of surface charge (Supplementary Fig. 5). Analysis of the electrostatic potential revealed that two prominent insertions of the sema domain, the extrusion and the insertion located between blades 1 and 2, form substantial charged patches. Both insertions are negatively charged in the class 1 semaphorins while in the class 2 semaphorins they form positively charged patches. Intriguingly, an area of the surface, corresponding to a co-receptor (neuropilin) binding site in the vertebrate Sema3s (ref. [30]), is distinguished by strong negative charge in the class 1 semaphorins (Supplementary Fig. 5). This surface has positively charged/neutral character in the class 2 semaphorins, similar to that in Sema3A. Neuropilin is not present in *Drosophila*, however, the distinctive characteristics of this area on the *Drosophila* semaphorin sema domains suggest that it could mediate class-specific co-receptor binding.

We also observed inter-class variations in the binding affinities of the *Drosophila* semaphorins to their plexin receptors. We generated a panel of the full-length ectodomains of semaphorin-1a (Sema1a$_{ecto}$), and semaphorin-1b (Sema1b$_{ecto}$), and full-length secreted semaphorin-2a (Sema2a$_{full}$) and semaphorin-2b (Sema2b$_{full}$). Sema1a$_{ecto}$ purified as a mixture of monomers and dimers, we therefore stabilized the dimer form by an Fc fusion.

In a microscale thermophoresis binding experiment, Sema2b$_{full}$ showed the tightest binding among all *Drosophila* semaphorins, interacting with PlexB$_{1-4}$ with an apparent K$_d$ of $3 \pm 3$ nM (Fig. 1f). The apparent affinity between Sema2a$_{full}$ and PlexB$_{1-4}$ was at least eight times less compared with that of Sema2b$_{full}$. A lower affinity was observed for class 1 semaphorins. Fc-tagged dimerized Sema1a$_{ecto}$ interacted directly with PlexA$_{1-4}$ giving an apparent K$_d$ of $141 \pm 52$ nM while monomeric Sema1b$_{ecto}$ bound PlexA$_{1-4}$ with a K$_d$ of $586 \pm 134$ nM (Fig. 1f).

**Determinants driving semaphorin specificity and promiscuity.** *Drosophila* class 1 semaphorins have been reported to bind PlexA while class 2 semaphorins have been shown to bind PlexB[12–14]. To confirm the specificity of *Drosophila* semaphorins to their cognate plexin receptors, we examined binding of class 1 semaphorins to PlexB and class 2 semaphorins to PlexA. As expected from the literature on semaphorin–plexin biology, we did not detect any measurable indication of PlexA or PlexB binding to class 2 or class 1 semaphorins, respectively (Supplementary Fig. 6e–h). Therefore plexin binding is semaphorin class-specific, but within class seems to be relatively promiscuous. To shed light on the molecular basis that determines semaphorin–plexin specificity, we identified and compared plexin-binding sites. As the semaphorin–plexin docking mode is conserved in all available complex structures, we mapped putative interaction interfaces on *Drosophila* semaphorins by superposition of the vertebrate Sema6A–PlexA2 complex[6]. The superposition revealed that the backbone architecture at the putative plexin binding site is nearly indistinguishable among *Drosophila* semaphorins. Thus, the semaphorin specificity is unlikely to be explained by differences in the shape of the binding site. Also, the observed semaphorin rigidity suggests that the specificity is not driven by conformational plasticity. Instead, we identified nine residue substitutions that are spatially spread throughout the putative binding site and show notable differences in binding chemistry (Fig. 1g). Notably, the potential molecular determinants of semaphorin class specificity for PlexA or PlexB include substitutions of hydrophobic residues with polar residues and vice versa at positions 120, 180, 223, 218, and 243 (considering Sema2a numbering) and further substitutions of charged versus uncharged residues at positions 217, 244, 251 and 252 (Supplementary Fig. 1). Conversely, there is a high degree of sequence conservation within classes at the putative binding site, in particular for the class 2 semaphorins.

**Class 1 is more divergent than class 2 semaphorins.** Sema2a and Sema2b are very similar and share over 67% sequence identity. A superposition of Sema2a$_{1-3}$ and Sema2b$_{1-3}$ crystal structures (Supplementary Fig. 3d) unveiled that Sema2a and Sema2b are also highly similar in structure with an rmsd of 1.1 Å over 608 Cα atoms (for Sema2a chain A and Sema2b chain A). Most notable variations are observed in the distribution of charged patches located on their surfaces (Supplementary Fig. 5). Contrary to the class 2 semaphorins, the ectodomain of Sema1a and Sema1b is structurally and sequentially more divergent (Fig. 1e and Supplementary Fig. 3e). The ectodomains share 43% sequence identity, and a structural superposition reveals an rmsd of 1.5 Å over 476 Cα atoms (for Sema1a chain A and Sema1b chain A). In the Sema1b ectodomain, the β2C–β2D loop is significantly extended while the β7A–β7B loop is extended in Sema1a (Supplementary Fig. 3b). These loops appear to be disordered as we

were only able to partially model them into the electron density. Sequence analysis revealed a substantial intraclass difference in the linker length between the PSI domain and the membrane. The linker in Sema1a is shorter by 35 residues compared with that of Sema1b. Intriguingly, we found that the cytoplasmic domain of Sema1b lacks the sequence previously determined to be crucial for reverse signalling of Sema1a[22,31], and therefore, Sema1b might be incapable of mediating reverse signalling. In addition, Sema1a and Sema1b differ markedly in the dimerization propensity of their ectodomains, as described in the next section.

**Sema1b is a monomer due to an amino acid substitution**. To characterise the oligomeric behaviour of the fly semaphorin extracellular regions, we produced the full-length ectodomains of the membrane attached *Drosophila* semaphorin-1a (Sema1a$_{ecto}$), and semaphorin-1b (Sema1b$_{ecto}$), and full-length secreted semaphorin-2a (Sema2a$_{full}$) and semaphorin-2b (Sema2b$_{full}$) in HEK293T cells. Multi-angle light scattering and SDS-PAGE analysis under reducing and non-reducing conditions (Fig. 2) revealed that both Sema2a$_{full}$ and Sema2b$_{full}$ exist as disulfide-linked dimers, Sema1a$_{ecto}$ is expressed as a mixture of a monomer and a disulfide-linked dimer, but Sema1b$_{ecto}$ appears to be a

monomer; no propensity to dimerize was observed up to a concentration of 2 mg/ml. Our crystal structures of *Drosophila* semaphorins revealed that the dimeric architecture in Sema1a, Sema2a and Sema2b is covalently secured by an intermolecular sema-to-sema disulfide bond formed by a cysteine from each of the β4B–β4 C loops of the opposing chains (Cys279 in Sema2a) (Supplementary Fig. 7a). A multiple sequence alignment of semaphorins indicates that the cysteine residue at this position is also conserved in the class 5 semaphorins 5A, 5B and 5c, in Sema4C and in *C. elegans* semaphorins cSMP2 (Supplementary Fig. 4). In Sema1b, the equivalent cysteine residue at position 254 is naturally substituted by a phenylalanine, and thus Sema1b fails to form the interchain disulfide bond as compared with Sema1a, Sema2a and Sema2b. When we mutated the phenylalanine 254 in Sema1b$_{ecto}$ to cysteine, we observed a mixture of a monomer and a disulfide-linked dimer similar to that of Sema1a$_{ecto}$ (Fig. 2). Conversely, mutation of the cysteine residue involved in dimerization in Sema1a$_{ecto}$ to serine completely abrogated dimerization, and no dimer was detected up to a concentration of 2 mg/ml. On the other hand, when we mutated the equivalent cysteine residue to serine in Sema2a$_{full}$ and Sema2b$_{full}$, both proteins partially maintained the homodimeric architecture and a mixture of a monomer and a non-covalent dimer was observed. What

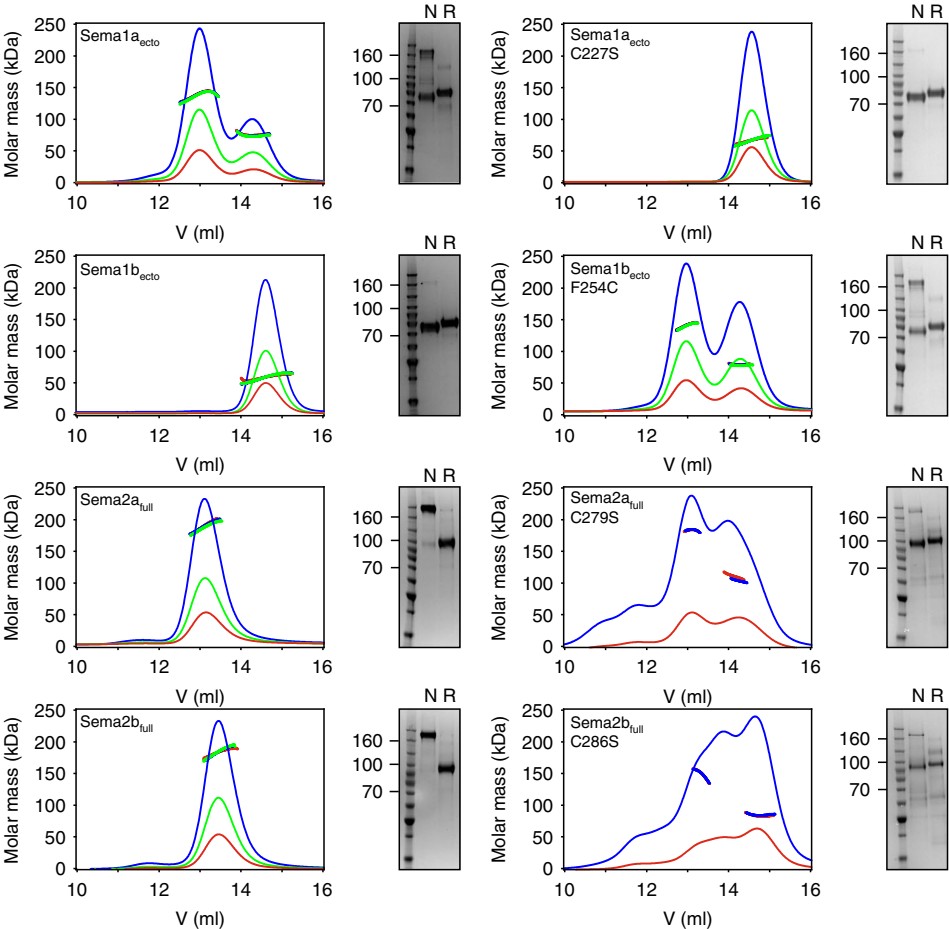

**Fig. 2** MALS and SDS-PAGE of *Drosophila* class1 and class 2 semaphorins. *Drosophila* semaphorins (left panel) were analysed using size-exclusion chromatography with multi-angle light scattering (MALS) and SDS-PAGE under reducing (R) and non-reducing (N) conditions. Sema1a$_{ecto}$ is a mixture of a monomer and a dimer in solution, and SDS-PAGE indicates that the dimer is disulfide-linked. Sema2a$_{full}$ and Sema2b$_{full}$ appear to be exclusive disulfide-linked dimers in solution. For Sema1b$_{ecto}$, MALS indicates an experimental molar mass of 75 ± 0.3 kDa, which is in agreement with the theoretical molar mass for a monomer (70 kDa). No peak shift towards higher molecular masses is observed at any of initial protein concentrations of 2.0 (blue), 1.0 (green), and 0.5 mg/ml (red). Mutation analysis (right panel) indicates that class 1 and 2 semaphorins differ in their propensity to dimerize non-covalently. Source data are provided as a Source Data file

determines these class-specific differences in dimerization propensity and why does Sema1b ectodomain fail to form at least a noncovalent dimer? We turned to analysis of our crystal structures of semaphorins to address this question.

**Class-specific differences in semaphorin homodimerization.** The overall dimeric architecture of Sema1a$_{ecto}$, Sema2a$_{full}$ and Sema2b$_{full}$ is similar to all previously reported semaphorin crystal structures[3,4,6–8]. In the class 2 semaphorins, besides the interchain disulfide bond, dimerization is mediated mainly by the top face of the sema domain using five protruding surface loops (Fig. 3a), which intertwine with each other to form a massive interface burying a total of 4496 Å$^2$. This interface consists of both polar and hydrophobic residues forming electrostatic and hydrophobic interactions. The Ig-like domain of the class 2 semaphorins is also involved in dimerization and contributes 21% to the total buried surface area of the interface. Remarkably, in the Sema2a structure, an N-linked glycan at residue N314 forms intermolecular interactions with the surface residues of the neighbouring chain (Fig. 4 and Supplementary Fig. 7b). Approximately 17% of the total buried surface area between the two chains of the dimers is contributed by this N-linked glycosylation. Of the Sema2a glycosylation sites, N314 (Asn-Cys-Ser) is the most conserved site considering all *Drosophila* and human semaphorins; it is present in all *Drosophila* semaphorins, and class 5 and class 6 vertebrate semaphorins (Supplementary Fig. 4). Previously reported crystal structures of class 6 semaphorins (Sema6A) were determined for deglycosylated proteins and the crystal structure of Sema1a reported here is at lower resolution and thus the glycans have been omitted from the model. However, this distinctive glycan is also well-ordered in the Sema1b and Sema2b structures. Taken together, we found class 2 semaphorins are exclusively dimers in solution and in the context of their crystal lattice, consistent with a strong propensity to dimerize. Their dimerization is mediated by the covalent interchain disulfide bond, and non-covalent interactions between the sema domains as well as Ig-like domains, also involving conserved N-linked glycan chains.

Conversely, the ectodomain of class 1 semaphorins shows a low propensity to dimerize non-covalently due to a lack of key elements stabilizing the dimeric architecture. First, class 1 semaphorins do not contain the Ig-like domain, and second, a number of residues that are conserved within the intertwined loops in class 2 are replaced in class 1 semaphorins by residues with different chemical nature (Fig. 3b–d). For example, charged residues Asp and Arg or Lys that form salt bridges between β5D–β6A loops in class 2 semaphorins are replaced by residues with non-charged side chains in class 1 semaphorins (Fig. 3). The low propensity to non-covalently dimerize and the natural substitution of cysteine 254 to phenylalanine together contribute to the monomeric state of Sema1b$_{ecto}$. Despite the low propensity to non-covalently dimerize, Sema1a$_{ecto}$ maintains the homodimeric architecture by reason of the covalent interchain disulfide bond.

**Sema2a and Sema2b can form a heterodimer.** Semaphorin heterodimerization has not been described to date. The *Drosophila* semaphorin crystal structures, along with previously reported mouse and human semaphorin crystal structures, show that the residues at the homodimeric interfaces vary significantly in sequence and chemical nature between semaphorin classes. Thus semaphorin heterodimerization between different semaphorin classes appears to be unlikely. However, our analyses prompted us to test whether intraclass heterodimerization is possible, as the level of residue conservation and similarity within a semaphorin class appears high enough (Supplementary Figs. 1, 4).

Furthermore, *Drosophila* Sema2a and Sema2b are good candidates for heterodimerization as they have been reported to have a similar expression pattern in the same tissue and at the same time[25].

To examine the ability of semaphorins to heterodimerize, we transiently co-transfected HEK293T cells with Sema2a and Sema2b constructs encoding TwinStrep and His6 tag, respectively. Five days post transfection, we observed Sema2a and Sema2b homodimers but also a Sema2a/2b heterodimer which we were able to pull down by two-step affinity chromatography (Fig. 5a, b). In solution, SEC-MALS measurements showed the purified Sema2a/2b heterodimer is a stable disulfide-linked dimer of 1:1 stoichiometry, which is clearly distinguishable from the homodimers by a double band on SDS-PAGE (Fig. 5c, d). The Sema2a/2b heterodimer also bound directly to PlexB$_{1–4}$ with an apparent K$_d$ of $13 \pm 4$ nM in a microscale binding experiment (Fig. 5e). The K$_d$ of the Sema2a/2b heterodimer lies between the apparent K$_d$ of Sema2a (25 nM) and Sema2b (3 nM).

We further investigated semaphorin heterodimerization between different semaphorin classes. We selected Sema1a and Sema2a as candidates for potential interclass heterodimerization as both Sema1a and Sema2a are disulfide-linked dimers that are formed by an intermolecular disulfide bridge located at the same position in the sema domain. To examine the ability of Sema1a and Sema2a to heterodimerize, we transiently co-transfected HEK293T cells with Sema1a$_{ecto}$ and Sema2a$_{full}$ constructs encoding His6 and TwinStrep tags, respectively. Using two-step affinity chromatography, we pulled down Sema1a$_{ecto}$ and Sema2a$_{full}$ homodimers, however, we were not able to detect the Sema1a/2a heterodimer (Supplementary Fig. 8) supporting our hypothesis that the heterodimerization between different semaphorin classes appears to be unlikely.

We also extended this study to vertebrate semaphorins. Similar to *Drosophila* class 2 semaphorins, mouse Sema3A and Sema3C are secreted proteins, and have been reported to form an intermolecular disulfide bond located in their C-terminal tail region[32,33]. Moreover, similar expression patterns for both Sema3A and Sema3C have also been reported for the same tissue at the same time[34]. Consistent with the previous experiment we found that mouse Sema3A and Sema3C form a stable disulfide-linked heterodimer of 1:1 stoichiometry (Fig. 5f, g).

## Discussion

Semaphorins are a large family of signalling molecules that elicit a wide range of responses in cells and tissues. Unlike the large and diverse vertebrate semaphorin family, the *Drosophila* semaphorin family consists of five semaphorins, which interact with two plexins. Class 1 and 2 semaphorins are best known for their role in neural circuit assembly. Here, we determined the crystal structures of class 2 semaphorins and the ectodomains of class 1 semaphorins. Overall, we found that *Drosophila* semaphorins show high levels of similarity in their structures. However, the crystal structures revealed several loops extending from the bottom face of the sema domain β-propeller, which are class specific and distinctive for *Drosophila* semaphorins. We showed that these loops in Sema1b have high intrinsic flexibility. Commonly, long and flexible loops can mediate protein recognition. Intriguingly, there is a precedent for a sema domain containing protein using this surface to mediate interaction; the extending loops in class 2 semaphorins align with the interaction interface between structurally similar MET receptor and its ligand HGF-β[29]. We also found substantial inter-class differences in charge distribution on the surfaces of *Drosophila* semaphorins. These differences are particularly apparent in the region equivalent to that used for neuropilin binding in mouse Sema3A[30]. Although class 1 and 2 semaphorins bind to different plexin receptors, the

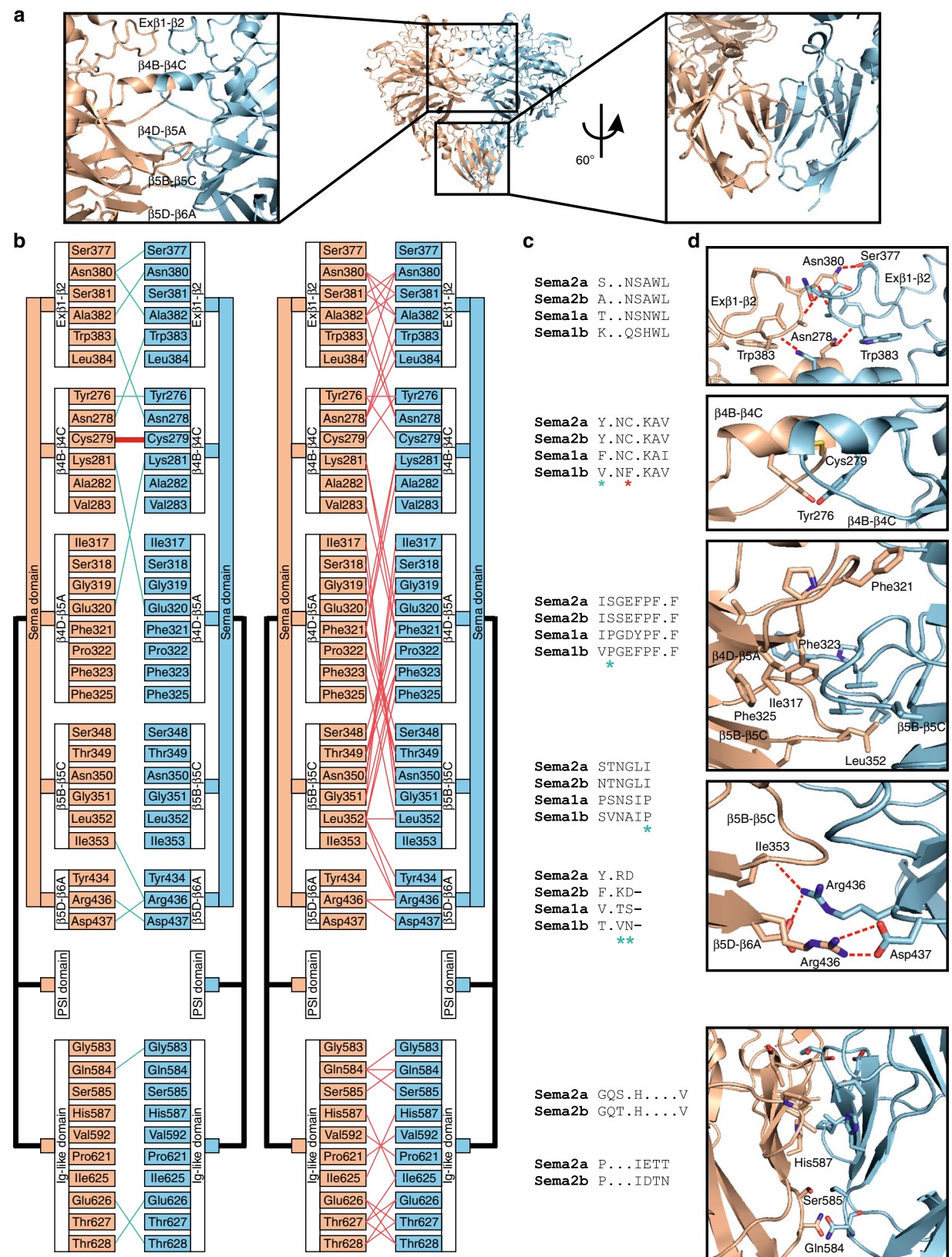

**Fig. 3** Protein–protein interaction at the Sema2a₁₋₃ homodimer interface. **a** Ribbon representation of the Sema2a₁₋₃ crystal structure with two zoom-in views showing a face-to-face interaction between the top surfaces of the sema domains formed by five protruding loops (left) and intermolecular interaction between Ig-like domains (right). **b** Details of intermolecular interactions depicted in (A). The left panel shows the interchain disulfide bond (red) and intermolecular hydrogen bonds while the right panel shows the other non-bonded contacts. **c** Sequence alignment of the residues involved in the Sema2a homodimerization from *Drosophila* semaphorins. The class-specific differences in propensity to dimerize can be explained by the class-specific residues that are shown by asterisks. **d** Close-up views of the Sema2a₁₋₃ homodimer interface

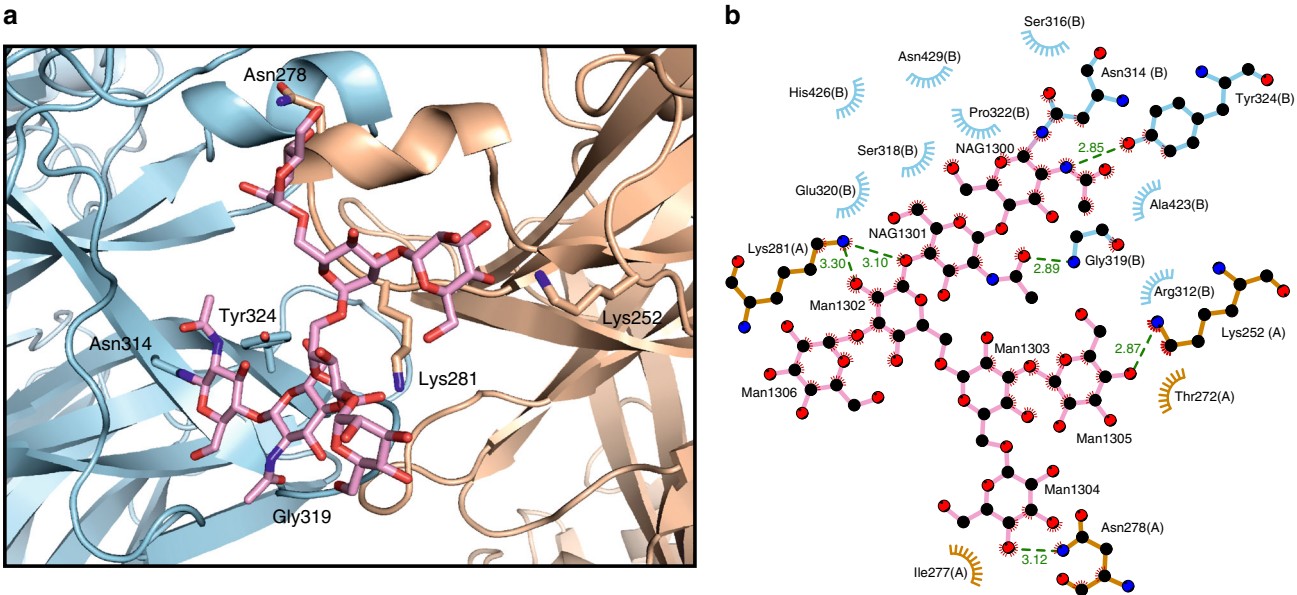

**Fig. 4** N-linked glycans are involved in the semaphorin homodimerization. **a** Close-up view of the N-linked glycans at residue N314 in the Sema2a$_{1-3}$ crystal structure. The N-linked glycans of chain B (blue) form intermolecular interactions with residues of chain A (orange). **b** Schematic representation of the interactions between the glycans at N314 and residues from chain A (blue) and chain B (orange) adapted from Ligplot[65]

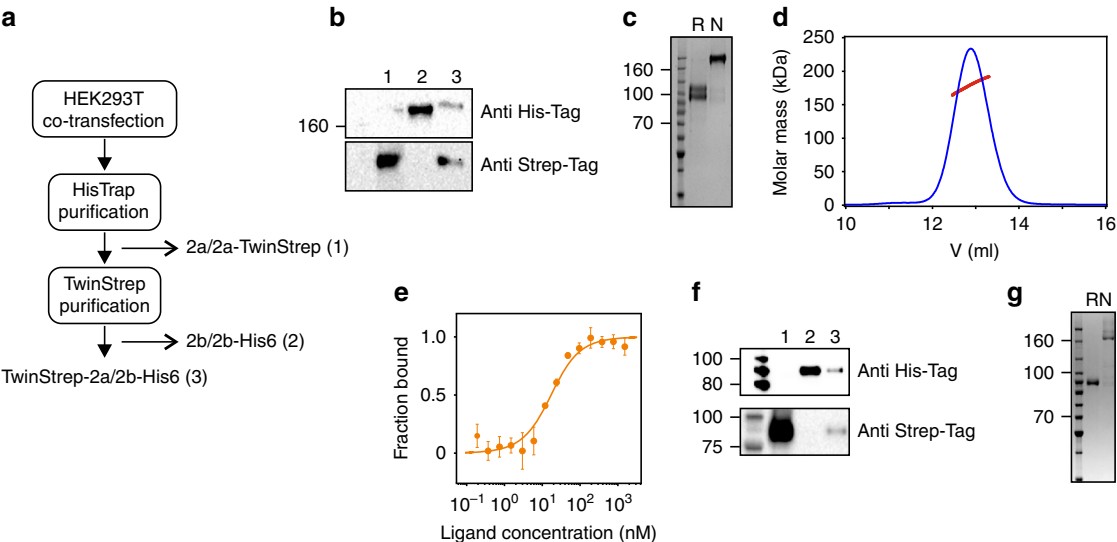

**Fig. 5** Semaphorin heterodimerization. **a** HEK293T cells were transiently co-transfected with constructs encoding Sema2a-TwinStrep tag and Sema2b-His6 tag. Five days post transfection, the Sema2a/2b heterodimer was purified by two-step affinity chromatography on a HisTrap column followed by TwinStrep tag purification on a StrepTactin column. The unbound homodimers in flow-through, Sema2a/2a (1) or Sema2b/2b (2), and the purified Sema2a/2b heterodimer (3) were collected and used for western blot analysis. Similarly, for mouse Sema3A/3 C heterodimer, constructs encoding Sema3A-His6 tag and Sema3C-TwinStrep tag were used and the Sema3A/3C heterodimer was purified as the *Drosophila* Sema2a/2b heterodimer. **b** Western blot analysis (SDS-PAGE under non-reducing conditions) of the unbound homodimers in flow-through (1) and (2), and the purified Sema2a/2b heterodimer (3). **c** SDS-PAGE of the purified Sema2a/2b heterodimer under reducing and non-reducing conditions shows that Sema2a/2b is a disulfide-linked heterodimer. **d** Size-exclusion chromatography with multi-angle light scattering of the Sema2a/2b heterodimer indicates an experimental protein mass of 156 ± 0.8 kDa, which is in agreement with the theoretical molar mass of 163 kDa. **e** Microscale thermophoresis binding experiment for PlexB$_{1-4}$-mVenus and the Sema2a/2b heterodimer. Error bars represent s.d. of three technical replicates. **f** Western blot analysis (SDS-PAGE under reducing conditions) of the unbound homodimers in flow-through (1) and (2), and the purified Sema3A/3 C heterodimer (3). **g** SDS-PAGE of the purified Sema3A/3 C heterodimer under reducing and non-reducing conditions. Source data are provided as a Source Data file

overall architecture of the putative plexin-binding site is almost identical between the classes. We identified nine potential hot-spot residues that are class specific and spread throughout the binding site. These residues probably determine the specificity of binding to either PlexA or PlexB. Binding experiments revealed

that class 2 semaphorins bind their plexin receptor, PlexB, with higher affinity than class 1 semaphorins bind their PlexA receptor. Within classes, Sema2b binds PlexB ~8 times more strongly than Sema2a. This difference may have functional relevance. Both Sema2a and Sema2b recognize the same plexin receptor but they

have been reported to mediate opposing effects in the embryonic central nervous system, Sema2a binding results in repulsion whilst Sema2b binding triggers attraction[14]. Given the essentially identical architecture of Sema2a and Sema2b, we conclude that their different binding properties either in strength or lifetime of binding to PlexB or interaction with additional co-receptor, determine their distinct functions.

We discovered that *Drosophila* Sema1a, Sema2a and Sema2b can be locked into a dimeric state through the formation of a sema-to-sema domain disulfide bond that is essential in the maintenance of dimer stability. This interface represents an alternative to the non-covalent interfaces reported before. We further found that Sema1b lacks this bond and our crystal structure determination and biophysical data in solution all show Sema1b$_{ecto}$ to be in a monomeric state due to the C254F substitution. This was an unexpected finding as disulfide bonds are strongly conserved among species and once disulfide bonds are acquired in proteins, they are rarely lost through evolution[35]. To date all crystal structures that include semaphorin sema-PSI regions have revealed them to conform to dimeric architectures[5]. The monomeric structure of Sema1b$_{ecto}$ we report here raises the intriguing possibility that semaphorin function may not be restricted to dimers, but also monomers could contribute to the complexity of biology mediated by the semaphorin family. This observation may be of particular relevance for semaphorins that lack interchain disulfide bonds.

In vertebrates, some, but not all, of the mammalian semaphorins can form interchain disulfide bonds at various points in their ectodomains (Fig. 6). Mouse Sema3A has been reported to form a disulfide-linked dimer using cysteines in the C-terminal tail[32,33], and this cysteine is highly conserved across all members of the mouse or human class 3 semaphorins. Also, human Sema4D has been shown to be a disulfide-linked dimer via cysteines located in the linker region between the Ig-like domain and transmembrane region[36], however, this cysteine is not conserved in the other members of the mouse or human class

4 semaphorins. Our sequence analysis indicates that the sema-to-sema domain disulfide bond, which we found in Sema1a, Sema2a and Sema2b, can potentially mediate dimerization in Sema4C and class 5 semaphorins. Human Sema7A has been reported to be a non-covalent dimer[8]. However, it might also be a disulfide-linked dimer because a single cysteine, which was not visible in the crystal structure, can be found in the linker between the Ig-like domain and the GPI anchor. For class 6 semaphorins, a construct encoding the sema and PSI domain of mouse Sema6A has been reported to form a monomer–dimer equilibrium in solution[6,7]. Previous structural studies on mouse Sema6A have revealed canonical sema-to-sema domain homodimerization mediated by non-covalent interactions[6,7]. However, as there is no cysteine in the linker connecting the PSI domain with the transmembrane region, mammalian class 6 semaphorins can potentially exist on the cell surface as a mixture of monomers and non-covalent dimers. Indeed, the dimerization propensity of these semaphorins may be weakened because the Ig-like domain, which has been shown to contribute to dimerization in Sema2a, Sema2b, Sema3A, Sema4D and Sema7A, is not present in class 6 semaphorin ectodomains. As reported here, the vertebrate class 6 semaphorins and invertebrate class 1 semaphorins are structural homologues. It is therefore intriguing that whilst *Drosophila* class 1 semaphorins may occur on the cell surface as a predominantly monomeric Sema1b and disulfide-linked dimeric Sema1a, their mammalian Sema6 homologues can potentially form an equilibrium of monomers and non-covalent dimers. The implications for biological function of semaphorin proteins in the monomeric state remains to be elucidated.

We further demonstrated that *Drosophila* class 2 and mammalian class 3 semaphorins could form heterodimers by direct interaction with members of the same semaphorin class. We also showed that the Sema2a/2b heterodimer binds PlexB. Overlapping expression, a prerequisite for biologically relevant heterodimerization, has been described for many members of the semaphorin family. Therefore, semaphorin heterodimerization

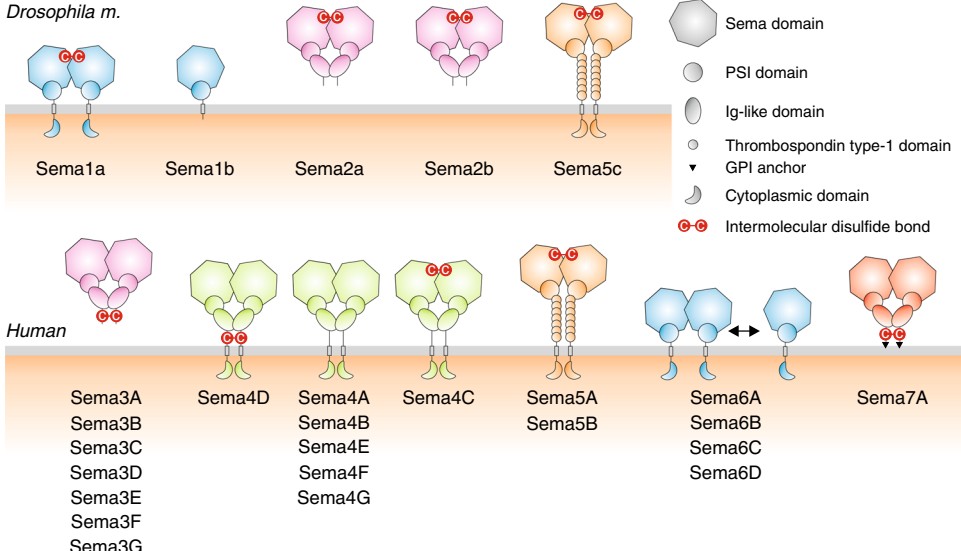

**Fig. 6** Oligomeric state of fly and human semaphorins. We found that the *Drosophila* Sema1a ectodomain and also full-length Sema2a and Sema2b are disulfide-linked dimers, while the Sema1b ectodomain is monomeric. Our sequence analysis indicates that the cysteine involved in the sema-to-sema disulfide bond is conserved in the class 5 semaphorins 5A, 5B and 5c, and in Sema4C. Sema3A and Sema4D have also been described as disulfide-linked dimers with the interchain disulfide bond located in a region followed by the Ig-like domain. This cysteine is highly conserved in all class 3 semaphorins, however, it is not conserved in the other class 4 semaphorins (Sema4A, 4B, 4E, 4F and 4G), in which the dimeric architecture is potentially stabilized by non-covalent interactions between the sema and Ig-like domains. It is not clear whether Sema7A is a non-covalent dimer or a disulfide-linked dimer. The Sema6A ectodomain has been reported as a mixture of a monomer and a non-covalent dimer in solution[6,7]. Remarkably, there is an obvious similarity with their structural homologues, *Drosophila* class 1 semaphorins

plausibly provides a mechanism allowing cross talk between receptors and co-receptors that can serve as an additional level of fine-tuning of cell signalling, as is for example observed in the disulfide-linked, dimeric signalling domains of the TGFβ/BMP pathway[37]. Notably, semaphorin heterodimers could bring together different classes of plexin receptors or neuropilins as co-receptors, which homodimers cannot because of their limited specificity. The mechanism to generate additional ligand diversity has a potentially greater impact in vertebrates because of the larger membership of the semaphorin classes.

Previous studies on semaphorins highlighted that despite a limited number of semaphorins, their functional complexity and versatility in establishing neuronal networks is enormous. In the canonical paradigm, semaphorins have been thought to function as homodimers, as dimerization of their plexin receptors is essential for triggering signalling. Perhaps, this model is too simplistic to explain the functional flexibility of these molecules. Monomeric semaphorins and semaphorin heterodimers can help us better understand how these molecules achieve their complexity and functional versatility.

## Methods

**Protein production**. Constructs encoding *Drosophila melanogaster* Sema1a$_{ecto}$, Sema1a$_{1-2}$, Sema1b$_{ecto}$, Sema1b$_{1-2}$, Sema2a$_{full}$, Sema2b$_{full}$, Sema2a$_{1-3}$, Sema2b$_{1-3}$, and PlexA$_{1-4}$ (residues 21N-606E, 21N-602Q, 37D-659S, 37D-602S, 26D-724V, 34D-736V, 27Y-671K, 33Y-679Q, and 28Q-730T, respectively) were cloned into the pHLsec vector in-frame with a C-terminal hexahistidine (His6) tag[38]. A construct encoding *D. melanogaster* PlexB$_{1-4}$ (residues 35E-730P) was cloned into the pBacPAK9 vector (Clontech) with an N-terminal GP64 signal peptide and C-terminal hexahistidine (His6) tag. A human IgGγ1 hinge and Fc-fusion construct of Sema1a (residues 21–602) was constructed using the pHL-FcHis vector[38]. For microscale thermophoresis (MST) experiments, PlexA$_{1-4}$ and PlexB$_{1-4}$ were C-terminally tagged with a monoVenus (mVenus) followed by the hexahistidine (His6) tag. For crystallization experiments, Sema1a$_{1-2}$, Sema1b$_{1-2}$, Sema2a$_{1-3}$ and Sema2b$_{1-3}$ were produced by transient transfection in either HEK293S-GnTI⁻ (ATCC CRL-3022) cells or HEK293T (ATCC CRL-3216) cells in the presence of the α-mannosidase inhibitor kifunensin[39]. For all other experiments, Sema1a$_{ecto}$, Sema1b$_{ecto}$, Sema2a$_{full}$, Sema2b$_{full}$, Sema1a$_{ecto}$-Fc and PlexA$_{1-4}$ were produced in HEK 293T cells without kifunensine. Sema1a$_{ecto}$, Sema1a$_{1-2}$, Sema1a$_{ecto}$-Fc, Sema1b$_{ecto}$, and Sema1b$_{1-2}$ were produced in HEK 293T cells maintained at 30 °C; all other constructs were produced at 37 °C. PlexB$_{1-4}$ was produced in High Five cells (*Trichoplusia ni*) (ATCC CRL-10859). The conditioned medium was collected 5 days post transfection (HEK293 cells) or 3 days post infection (High Five cells) and proteins were purified from buffer-exchanged media by immobilized metal-affinity (HisTrap FF column, GE Healthcare) and size-exclusion chromatography (Superdex 200 16/60 column, GE Healthcare).

For Sema2a/2b heterodimer production, a construct encoding *D. melanogaster* Sema2a$_{full}$ was cloned into pHLsec vector in-frame with a C-terminal TwinStrep tag and HEK293T cells were co-transfected with both Sema2a$_{full}$-pHLsec-TwinStrep and Sema2b$_{full}$-pHLsec-His6 tag. Similarly, for the mouse Sema3A/3C heterodimer production, mouse Sema3A (residues 26–730) and mouse Sema3C (residues 22–711) were cloned into the pHLsec vector in frame with the C-terminal 3C-Avi-His6 tag and the C-terminal TwinStrep tag, respectively. In addition, furin cleavage sites were mutated to prevent the cleavage in both Sema3A and Sema3C (Sema3A—R551, 555A and Sema3C—R548, 552A). For heterodimer production, 5 days post transfection the conditioned medium was collected and buffer exchanged using a QuixStand diafiltration system (GE Healthcare). Both Sema2a/2b and Sema3A/3C heterodimers were purified by immobilized metal-affinity chromatography using the HisTrap FF column (GE Healthcare) followed by affinity chromatography using a StrepTactin XT Superflow column (IBA) and further purified by size-exclusion chromatography using a Superdex 200 16/60 column (GE Healthcare).

Site-directed mutagenesis of *Drosophila* semaphorins was carried out by overlap-extension PCR, and the resulting PCR products were cloned into the pHLsec vector as described above. All mutant proteins were secreted at similar levels to the wild-type proteins. A list of all primers used in this study is shown in Supplementary Table 2.

**Protein crystallization**. Crystallization trials were set up using a Cartesian Technologies pipetting robot and consisted of 100 nl protein solution and 100 nl reservoir solution[40]. All crystals were grown at 20 °C in sitting drops vapour diffusion.

Sema1b$_{1-2}$ crystallized in 0.2 M trisodium citrate and 20% (w/v) PEG 3350. Sema1a$_{1-2}$ crystallized in 0.1 M Tris-HCl (pH 8.0), 0.2 M ammonium sulfate and 25% (w/v) PEG 3350. Sema2a$_{1-3}$ crystals grew in 0.1 M Bis-Tris (pH 6.5) and 20%

(w/v) PEG 5000 MME. Sema2b$_{1-3}$ crystallized in 0.1 M trisodium citrate (pH 5.0) and 10% (w/v) PEG 6000. Crystals were cryoprotected by soaking in reservoir solution supplemented with 25% (v/v) glycerol for both Sema1a$_{1-2}$ and Sema1b$_{1-2}$, and 25% (v/v) ethylene glycol for both Sema2a$_{1-3}$ and Sema2b$_{1-3}$, and then flash-cooled in liquid nitrogen.

**Data collection, structure determination and analysis**. Diffraction data for Sema1a$_{1-2}$ and Sema1b$_{1-2}$ were collected at 100 K at Diamond Light Source beamline I03 and indexed, integrated and scaled using the automated XIA2[41], XDS[42] and XSCALE[42]. Anisotropy correction was performed using the STAR-ANISO web server (http://staraniso.globalphasing.org/cgi-bin/staraniso.cgi). Anisotropy correction of Sema1a$_{1-2}$ yielded an ellipsoidal resolution boundary with limits of 4.2, 4.2 and 3.2 Å along the $a^*$, $b^*$ and $c^*$ axes, respectively, while anisotropy correction of Sema1b$_{1-2}$ yielded an ellipsoidal resolution boundary with limits of 2.7, 2.7 and 3.5 Å along the $a^*$, $b^*$ and $c^*$ axes, respectively. Crystals of Sema1a$_{1-2}$ were merohedrally twinned via three twin operators -h, -k, -l (twin fraction 0.277); h, -h -k, -l (twin fraction 0.107); -k, -h -l (twin fraction 0.099) (calculated with Phenix-Xtriage). The structure of Sema1b$_{1-2}$ was solved by molecular replacement in PHASER[43] with the Sema6A structure (pdb 3oky)[6], as the search model. The structure of Sema1a$_{1-2}$ was initially solved by molecular replacement in PHASER[43] using the structure of Sema1b$_{1-2}$. The partial models for both Sema1a$_{1-2}$ and Sema1b$_{1-2}$ were rebuilt automatically by BUCCANEER[44] and completed by several cycles of manual rebuilding in COOT[45]. The resultant model of Sema1a$_{1-2}$ was improved with MR-Rosetta[46]. Sema1b$_{1-2}$ was refined in Buster[47] and PHENIX[48]. For refinement of Sema1a$_{1-2}$, we performed twin refinement in the Phenix package with individual twin operators. Application of the first operator (-h, -k, -l) provided the best overall refinement statistic. We were not able to use multiple twin operators in the Phenix refinement because currently, Phenix supports only a single twin operator. We also performed twin refinement using multiple twin operators in Refmac[49]; however, the overall refinement statistic resulting from this procedure in Refmac was not as good as when using the first single operator only in Phenix. The electron density maps resulting from the two procedures were essentially identical. Thus, for simplicity, we report the results using the first operator for twin refinement of Sema1a in Phenix.

Diffraction data for Sema2a$_{1-3}$ were collected at 100 K at European Synchrotron Radiation Facility (ESRF) beamline BM14 and indexed, integrated and scaled using HKL2000 suite[50]. The structure of Sema2a$_{1-3}$ was determined by molecular replacement in PHASER[43] using the structure of Sema4D (pdb 1olz)[3], as the search model. This partial model was rebuilt automatically by ARP/wARP[51] and completed by several cycles of manual rebuilding in COOT[45] and refinement in PHENIX[48].

Diffraction data for Sema2b$_{1-3}$ were collected at 100 K at European Synchrotron Radiation Facility (ESRF) beamline ID23–1 and indexed, integrated and scaled using the HKL2000[50] suite. The structure of Sema2b$_{1-3}$ was determined by molecular replacement in PHASER[43] using the structure of Sema2a$_{1-3}$, as the search model. This solution was completed by model building in COOT[45] and refinement in PHENIX[48]. All models were validated with MolProbity[52]. Data collection and refinement statistics are given in Supplementary Table 1. Ramachandran statistics are as follow (favoured/disallowed (%)): Sema1a$_{1-2}$ 94.98/0, Sema1b$_{1-2}$ 96.27/0, Sema2a$_{1-3}$ 97.08/0 and Sema2b$_{1-3}$ 96.27/0. Alignments were generated with Clustal Omega[53], structural alignment was performed using PDBeFold[54], the structure-based phylogenic tree was calculated with SHP[55], buried surface areas of protein–protein interactions were calculated with PISA[56], and electrostatics potentials were generated with APBS[57]. Figures were produced with PyMOL (Schrodinger, LLC), ESPRIPT[58] and Corel Draw (Corel Corporation).

**SEC-MALS**. Proteins were injected onto the Superdex 200 Increase 10/300 column (GE Healthcare) at a flow rate of 0.5 ml/min in 15 mM HEPES (pH 7.4) and 150 mM NaCl. The SEC column was coupled with a static light-scattering (DAWN HELEOS II, Wyatt Technology), differential refractive index (Optilab rEX, Wyatt Technology) and Agilent 1200 UV (Agilent Technologies) detectors. The molecular mass of glycoproteins containing N-linked oligomannose-type sugars was determined using an adapted RI increment value (dn/dc standard value, 0.185 ml/g). Data were analysed using the ASTRA software (Wyatt Technology).

**Microscale thermophoresis (MST)**. MST measurements were performed using a Nanotemper Monolith NT.115 instrument (Nanotemper) at 25 °C in 15 mM HEPES (pH 7.4), 150 mM NaCl, 2 mM CaCl$_2$ and 0.005% (v/v) Tween-20. To determine the binding affinity between *Drosophila* plexins and semaphorins, a dilution series was prepared. A concentration of the plexin fused with fluorescent protein mVenus was kept constant in all samples and the unlabelled semaphorin was varied in 1:1 dilution to give a titration. The samples were incubated 1 h at room temperature before filling into the standard capillaries (Nanotemper). The LED power was set to 40%. To find the best thermophoretic setting, a measurement at 20, 40, 60 and 80% MST power was performed and the best signal to noise ratio was obtained by using 80% MST power. The overall measurement time consisted of 5 s of cold fluorescence followed by IR-laser on and off times set at 30 and 5 s. Data were analysed with the MO Affinity Analysis v2.1.3 software (Nanotemper). The experiments were replicated thrice.

**Molecular dynamics simulations**. Molecular dynamics simulations of Sema1b$_{1-2}$ and Sema2a$_{1-3}$ were performed in Gromacs[59] using the AMBER99SB-ILDNP* force field[60,61]. The missing residues in Sema1b$_{1-2}$ were modelled in Modeller[62,63]. Before the simulation, the protein was immersed in a box of SPC/E water, with a minimum distance of 1.0 nm from the box edge. A total of 150 mM NaCl was added using genion. Long-range electrostatics were treated with the particle-mesh Ewald summation[64], and bond lengths were constrained using the P-LINCS algorithm. The integration time step was 5 fs. The v-rescale thermostat and the Parrinello–Rahman barostat were used to maintain a temperature of 300 K and a pressure of 1 atm. Simulations were carried out in triplicates of 100 ns each. The system was energy minimized using 1000 steps of steepest descent and equilibrated for 200 ps with restrained protein heavy atoms. Snapshots were extracted every 500 ps from each trajectory.

**Western blotting**. Proteins were separated by NuPAGE 4–12% Bis-Tris gels (ThermoFisher Scientific) and transferred to nitrocellulose membranes (Amersham Protran Premium, 0.45 μm). The membranes were blocked with 5% nonfat dry milk (His6 tagged proteins) or 5% BSA (TwinStrep tagged proteins) in PBS for 3 h at room temperature. For His6 tag detection, the membranes were incubated with primary antibody (Penta-His Antibody, Qiagen, cat. no. 34660, dilution 1:1000) for 1 h at room temperature, washed three times for 10 min with PBS and incubated for 1 h at room temperature with secondary antibody conjugated to horseradish peroxidase (Anti-mouse IgG peroxidase polyclonal goat antibody, Sigma, cat. no. A0168, dilution 1:10,000). For TwinStrep tag detection, the membranes were incubated with antibody conjugated to horseradish peroxidase (Precision Protein StrepTactin-HRP Conjugate, BioRad, cat. no. 1610380, dilution 1:1000) for 1 h at room temperature. After washing three times for 10 min with PBS, the signal was detected using ECL (BioRad). Uncropped images of all western blots are shown in the Source data file.

**Reporting summary**. Further information on research design is available in the Nature Research Reporting Summary linked to this article.

## Data availability
Structure factors and coordinates have been deposited in the Protein Data Bank with identification numbers PDB: 6QP9, 6FKK, 6QP7 and 6QP8. Source data underlying Figs. 1f, 2, 5b, 5c, 5e–g, and Supplementary Figs. 2, 6e–h and 8 are available as a Source Data file. Other data and materials are available upon request from the corresponding authors.

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

## Acknowledgements

We thank the staff of Diamond Light Source and ESRF for support and access to beamlines I03, I24 and BM14, respectively; Weixian Lu for help with tissue culture; Thomas Walter for crystallization technical support; David Staunton for assistance with biophysical experiments. The work was funded by Cancer Research UK (C375/A17721 to E.Y.J., and C20724/A14414 and C20724/A26752 to C.S.), the Medical Research Council UK (MR/M000141/1 to E.Y.J) and the European Research council (647278 to C.S.). The Wellcome Centre for Human Genetics is supported by Wellcome Trust Centre grant 203141/Z/16/Z. D.R. and V.J. were supported by EMBO Long-Term Fellowships (ALTF 604–2014) and (ALTF 1061–2017), respectively.

## Author contributions

D.R., C.S. and E.Y.J designed the study. D.R., T.M. and R.A.R produced proteins. D.R., R.A.R., C.S. and K.H. collected data and solved crystal structures, D.R. performed biophysical measurements. D.R. and M.R. carried out molecular dynamics simulations. D.R. and V.J produced semaphorin heterodimers. D.R. and E.Y.J. wrote the paper with input from all authors.

## Additional information

**Competing interests:** The authors declare no competing interests.

**Peer Review Information**: *Nature Communications* thanks Rob Meijers and the anonymous reviewers for their contribution to the peer review of this work. Peer reviewer reports are available.

