## [Peer Review File · Nature Communications]

Reviewers' comments:

Reviewer #1 (Remarks to the Author):

The manuscript of Rozbesky and Jones presents result in crystallography to obtain the not yet resolved structures of drosophila semaphorins. The accompanying in vitro experiments substantiate the differences between the semaphorins in that some are disulphide bonded dimers, whereas others are not. Furthermore, heterodimerization appears to be possible between members of each sema class but not between them. Overall, as presented, the paper is focused/concise but is a brief report of more interest to a specialized audience. Specifically, the authors miss the opportunity to build on their previous work on sema/plexin clustering and this reviewer wonders whether a detailed analysis of crystal contacts is warranted (even though the structures are in different crystal forms). Especially when it comes to cross signaling, the role of cis-interactions has already been demonstrated, e.g. most recently <https://www.ncbi.nlm.nih.gov/pubmed/30827914>. Also, a reference to, if not engagement with a recent analysis of the evolution of plexins and semaphorins would be very interesting <https://www.ncbi.nlm.nih.gov/pubmed/30760850>.

The authors have used the mutagenesis of glycosylation sites in clever ways to confirm structures and this is hinted in Fig. 4; this reviewer wonders whether an analysis of glycosylation sites in the semaphorins studied would add another dimension to the paper.

A majority of inferences from the structural/in vitro analyses remain speculative without cell based assays, in this case, involvement of the plexin/sema Drosophila community would be needed to raise the paper to the level where the structures provide confirmatory insights.

Finally, this reviewer notices that few of the other plexin/sema structural group's work is cited, not even Siebold's and Jone's own 2013 review.

Reviewer #2 (Remarks to the Author):

The manuscript by Rozbesky et al. on the structural and biophysical characterization of Drosophila semaphorins (Semas) provides an important contribution to the mechanistic understanding of how

these important guidance molecules function. Several crystal structures of different classes of *Drosophila* Semas are determined, and a systematic analysis of how these molecules dimerize is presented for the first time. The authors show that homodimerization can occur covalently through a disulphide bond, or non-covalently through domain interactions. Interestingly, the authors also show that heterodimerization between Semas belonging to the same class can occur, leading to the realization that different oligomerization scenarios may contribute to functional fine-tuning of semaphoring/plexin interactions.

I have one serious concern regarding the only kinetics experimental technique (MST presented in this manuscript, and a minor concern regarding the crystallographic analysis of Sema1a. Otherwise, I think this is a very interesting study and I strongly recommend publication in *Nature Communications*.

Please find my detailed comments below:

1. The MST experiments are done by keeping the concentration of Plexin constant, while varying the concentrations of different labeled Semas. A large difference is seen for Sema1b, which is the only monomeric construct tested in MST. A general problem with MST is that it does not measure protein-protein interactions directly, but rather measures a change in the diffusion rate of the labeled protein. It is therefore entirely possible that the shift in diffusion rates of labeled Sema is not dependent on the presence of Plexin, but only on the concentration of Sema. The authors should do a control experiment, and do the same concentration series of all the Semas without the Plexin ligand being present, to show that the change in diffusion rate is not due to changes in Sema conformation alone.
2. The crystal structure of Sema1a is based on a low resolution X-ray data set (to 3.6 Angstrom). First of all, I am surprised that the anisotropic correction does not yield a higher resolution data set, which to me indicates that the space group may not be correct. More importantly, the authors report that the X-ray data are merohedrally twinned with a single twin operator, based on an analysis by the Xtriage program from the Phenix package. However, the attached PDB Validation report on the Sema1a structure reports an Xtriage analysis with 3 twin operators. Could the authors please provide a more detailed analysis of the X-ray data set to confirm that the space group is indeed correct, and the twin operator(s) are not a remnant from a different crystal or pseudo-symmetry?
3. It is not entirely clear what the MD simulations contribute to this study. The claim that “there is no significant difference in the dynamic behavior between monomeric Sema1b and dimeric Sema2a” is surprising, since the disulphide bond that dimerizes Sema2a will allow substantial conformational reorientations of the individual Sema2a molecules. I am not sure it is worthwhile to include these simulations in this study.

Reviewer #3 (Remarks to the Author):

The manuscript by Jones and colleagues is a highly interesting and important paper. In particular, Semaphorins (Semas) are the largest family of guidance cues (with over 20 family members conserved from invertebrates to vertebrates), playing multiple different critical roles in most, if not all tissues. Yet, we still have a poor understanding of how Semaphorins work, the interaction/integration of different semaphorins, and how the different Semaphorins work to activate their Plexin receptors. Here, the authors using the important *Drosophila* model, determined crystal structures for all members of *Drosophila* class 1 and 2 Semaphorins (i.e., 4 out of the 5 Semaphorins present in *Drosophila*). This strategy was very fruitful and among a range of important observations on the Semaphorins that emerged from the structure – their work revealed that Semas use a disulphide bond to stabilize dimer formation of Sema domains. The authors structural and biophysical data also suggest that one of these Semaphorins, Sema1b, is a monomer, indicating that semaphorin function appears not to be restricted to dimers. The authors also, for the first time, demonstrate that semaphorins are able to form heterodimers with members of the same semaphorin class. These are also very interesting results since heterodimerization provides a potential further mechanism to allow for fine-tuning of Semaphorin function and cell-cell signaling.

In total, therefore, this is significant work and will be of high-value to the multidisciplinary community. Further, the authors have done a nice job of assembling the work – nicely combining structure, biophysical approaches, and biochemistry. The writing and data presentation is also clear with a lot of great extras and insights – including supporting heterodimerization of Semas for the first time. And thus, while there is no functional/physiological testing of their structural/biochemical findings, the manuscript stands on its own (with all the data from the 4 structures) – and thus I believe functional/physiological studies go beyond the scope of the present study and should await future analysis.

However, with that being said, I believe a few additional experiments would complete and add to the value of the manuscript (and the authors have the reagents already in hand to address these questions):

1) Related to Fig. 1F, was binding seen between Class 1 Semas and PlexB or Class 2 Semas and PlexA? This would be important to test. The authors have the protein and the data should be presented in Suppl Figures. Especially since the authors say “*Drosophila* class 1 semaphorins have been reported to bind PlexA while class 2 semaphorins has been shown to bind PlexB11-13. Therefore plexin binding is semaphorin class-specific, but within class seems to be relatively promiscuous.” This is also very important based on Fig. 1G and the authors statement “Notably, the potential molecular determinants of semaphorin class specificity for PlexA or PlexB include substitutions of hydrophobic

residues with polar residues and vice versa at positions 120, 180, 223, 218, and 243 (considering Sema2a numbering) and further substitutions of charged versus uncharged residues at positions 217, 244, 251 and 252 (Fig. S1).” And also that “Conversely, there is a high degree of sequence conservation within classes at the putative binding site, in particular for the class 2 semaphorins.”

2) The authors demonstrate that semaphorins can form heterodimers with members of the same semaphorin class. What about between classes (such as the ecto domain of Sema-1a and Sema-2a or Sema-2b)? For similar reasons as noted above for 1), these two different possibilities should be tested (and also whether Sema-1aEcto and Sema-1bEcto can dimerize), and the authors have the tools in hand.

3) The authors say “Despite the low propensity to non-covalently dimerize, Sema1aecto maintains the homodimeric architecture by reason of the covalent interchain disulphide bond.” Yet, it looks like the class 6 semas (which would be vertebrate orthologues of class 1 Semas) do not have the Cys for dimerization (Fig. S4). How do they dimerize? There is a little discussion of this but this should be expanded in the discussion. Also, on a related note, the authors should comment on whether some caution related to the strong conclusions (in the abstract, intro, discussion, and Fig. 6) on Sema1b and its working as a monomer might be warranted. For example, since Class 1 Semas are transmembrane (and are likely to function as transmembrane ligands in vivo), might there be roles for the transmembrane or intracellular regions of Sema-1b in dimerization (i.e., can the authors exclude that)?

Minor:

1) The title for the legend of Fig 1 states “Crystal structure of Drosophila Semaphorins”. However, since one the Drosophila Semaphorins, Sema5c, is not included, the authors should change the title to “Crystal structures of Class 1 and Class 2 Semphorins” to not be misleading.

2) Lines 142-144: “In class 1 semaphorins, two distinct loops with no obvious structural and functional role are substantially extended from the bottom face of the β -propeller, β 5A- β 5B and β 7A- β 7B in Sema1a, and β 2C- β 2D and β 6C- β 6D Sema1b (Fig S3A).” This sentence refers to Fig S3B, not Fig S3A, and so should be corrected.

3) Lines 153-155: “Intriguingly, an area of the surface, corresponding to a co-receptor (neuropilin) binding site in the vertebrate Sema3s (ref 27), is distinguished by strong negative charge in the class 1 semaphorins (Fig. S5).” While this region is nicely mapped in yellow in the figure (Fig. S5), for the general reader, the authors should indicate here in the main text if and how this differs from the electrostatic potential of the neuropilin binding site in the vertebrate Sema3s. This is particular

interesting since neuropilin is not present in *Drosophila*, but a related protein could serve the same function.

4) Lines 162 -164 “In a microscale thermophoresis binding experiment, Sema2bfull showed the tightest binding among all *Drosophila* semaphorins, interacting with PlexB1-4 with an apparent K_d of 3 nM 164 (Fig. 1G).” This sentence refers to Fig 1F, not Fig 1G, and so should be corrected.

5) A lower affinity was observed for class 1 semaphorins. Fc tagged dimerized Sema1aecto interacted directly with PlexA1-4 giving an apparent K_d of 141 nM while monomeric Sema1becto bound PlexA1-4 with a K_d of 586 nM.” This sentence refers to Fig. 1F and so that should be included at the end of the sentence.

Point-by-point response to the reviewers' comments

Manuscript: NCOMMS-19-07496

Title: Diversity of oligomerization in *Drosophila* semaphorins suggests a mechanism of functional fine-tuning

Authors: Daniel Rozbesky, Ross A. Robinson, Vitul Jain, Max Renner, Tomas Malinauskas, Karl Harlos, Christian Siebold, and E. Yvonne Jones

June 24, 2019

We thank the reviewers for their interest in our work and constructive comments. Alongside guidance from the Editor these comments have informed our revision and we feel have resulted in an improved manuscript.

Reviewer #1:

The manuscript of Rozbesky and Jones presents result in crystallography to obtain the not yet resolved structures of drosophila semaphorins. The accompanying in vitro experiments substantiate the differences between the semaphorins in that some are disulphide bonded dimers, whereas others are not. Furthermore, heterodimerization appears to be possible between members of each sema class but not between them. Overall, as presented, the paper is focused/concise but is a brief report of more interest to a specialized audience. Specifically, the authors miss the opportunity to build on their previous work on sema/plexin clustering and this reviewer wonders whether a detailed analysis of crystal contacts is warranted (even though the structures are in different crystal forms). Especially when it comes to cross signaling, the role of cis-interactions has already been demonstrated, e.g. most recently <https://www.ncbi.nlm.nih.gov/pubmed/30827914>. Also, a reference to, if not engagement with a recent analysis of the evolution of plexins and semaphorins would be very interesting <https://www.ncbi.nlm.nih.gov/pubmed/30760850>.

We agree with the reviewer that analyses of crystal packing can be very informative for revealing interaction/oligomerization propensities. We have indeed thoroughly analyzed the crystal packing of our semaphorin crystal structures and have failed to identify any recurrent interfaces or interaction modes of potential biological significance other than dimer formation. We now note this in the revised text (page 5). *Drosophila* Sema1a, Sema2a and Sema2b crystallized as disulphide-linked dimers, and we have confirmed their dimeric architecture by SEC-MALS analysis and SDS-PAGE analysis under reducing and non-reducing conditions (Fig. 2). Our crystal packing analysis of Sema1b revealed that Sema1b forms five contacts with symmetry-related molecules; however, the lattice packing provides no evidence for Sema1b dimerization similar to that observed for all previously determined semaphorin structures, nor for any other mode of association.

We note the reviewer's comments on two recent publications and now include references to these papers in the revised manuscript (please see response to comment below).

The authors have used the mutagenesis of glycosylation sites in clever ways to confirm structures and this is hinted in Fig. 4; this reviewer wonders whether an analysis of glycosylation sites in the semaphorins studied would add another dimension to the paper.

We thank the reviewer for their kind comment on our previous analyses of glycosylation sites and agree that such analyses can be very informative. Our crystal structures of *Drosophila* semaphorins

show a high level of N-linked glycosylation and we have added text to introduce this point early in the text, in our general description of the crystal structures (page 5), and to alert the reader to our discussion of a distinctive site later in the text. The N-linked glycosylation site at N314 (Sema2a numbering) is highly conserved across classes and species and we discuss the role of this glycan in the text on page 10.

A majority of inferences from the structural/in vitro analyses remain speculative without cell based assays, in this case, involvement of the plexin/sema *Drosophila* community would be needed to raise the paper to the level where the structures provide confirmatory insights.

We respectfully submit that such experiments are beyond the remit of the present paper.

Finally, this reviewer notices that few of the other plexin/sema structural group's work is cited, not even Siebold's and Jones's own 2013 review.

We have added all the references suggested to the main text (Stedden CG et al. *Curr Biol* 2019 – Page3; Alves CJ et al. *Sci Rep* 2019 – Page5, Siebold & Jones. *Semin Cell Dev Biol* 2013 – Page2).

Reviewer #2:

The manuscript by Rozbesky et al. on the structural and biophysical characterization of *Drosophila* semaphorins (Semas) provides an important contribution to the mechanistic understanding of how these important guidance molecules function. Several crystal structures of different classes of *Drosophila* Semas are determined, and a systematic analysis of how these molecules dimerize is presented for the first time. The authors show that homodimerization can occur covalently through a disulphide bond, or non-covalently through domain interactions. Interestingly, the authors also show that heterodimerization between Semas belonging to the same class can occur, leading to the realization that different oligomerization scenarios may contribute to functional fine-tuning of semaphoring/plexin interactions.

I have one serious concern regarding the only kinetics experimental technique (MST presented in this manuscript, and a minor concern regarding the crystallographic analysis of Sema1a. Otherwise, I think this is a very interesting study and I strongly recommend publication in *Nature Communications*.

Please find my detailed comments below:

1. The MST experiments are done by keeping the concentration of Plexin constant, while varying the concentrations of different labeled Semas. A large difference is seen for Sema1b, which is the only monomeric construct tested in MST. A general problem with MST is that it does not measure protein-protein interactions directly, but rather measures a change in the diffusion rate of the labeled protein. It is therefore entirely possible that the shift in diffusion rates of labeled Sema is not dependent on the presence of Plexin, but only on the concentration of Sema. The authors should do a control experiment, and do the same concentration series of all the Semas without the Plexin ligand being present, to show that the change in diffusion rate is not due to changes in Sema conformation alone.

We agree that MST does not measure protein-protein interactions directly but rather changes in the diffusion rate of the labelled sample. Initially we wanted to use standard methods for the analysis of protein-protein interactions; however, we were not able to use surface plasmon resonance (SPR)

binding assays because all *Drosophila* semaphorin and plexin molecules showed high non-specific binding onto the surface of CM5 or SA sensor chips. Also, we were not able to use ITC (isothermal titration calorimetry) measurements because *Drosophila* semaphorins and plexins were not expressed in quantities sufficient for ITC.

With regard to the MST experiments, we believe that the reviewer's comment may be based on a partial misunderstanding, namely that the "The MST experiments are done by keeping the concentration of Plexin constant while varying the concentrations of different labelled Semas." In our MST experiment, the concentration of the plexin fused with fluorescent protein mVenus was kept constant in all samples, and the unlabeled semaphorin was varied in 1:1 dilution to give a titration. Thus, in this experimental setup, the shift in diffusion rates of fluorescent plexins (with a constant concentration in all capillaries) depends on the concentration of unlabelled semaphorins, which were used in dilutions. Furthermore, before each MST measurements, we performed capillary scans to check fluorescence intensity and the quality of fluorescently labelled molecules. In all experiments, we observed a symmetrical fluorescence peak demonstrating that the fluorescent samples were of good quality and showed no non-specific binding to the surface of capillaries. For each measurement, the fluorescence intensity was optimal for MST experiments (between 200-2500 fluorescent counts) and did not vary more than 10% between different capillaries. In response to the reviewer's concerns, we have added representative capillary scans to Supplementary Fig. 6 to further demonstrate the quality of the MST measurements.

2. The crystal structure of Sema1a is based on a low resolution X-ray data set (to 3.6 Angstrom). First of all, I am surprised that the anisotropic correction does not yield a higher resolution data set, which to me indicates that the space group may not be correct. More importantly, the authors report that the X-ray data are merohedrally twinned with a single twin operator, based on an analysis by the Xtrige program from the Phenix package. However, the attached PDB Validation report on the Sema1a structure reports an Xtrige analysis with 3 twin operators. Could the authors please provide a more detailed analysis of the X-ray data set to confirm that the space group is indeed correct, and the twin operator(s) are not a remnant from a different crystal or pseudo-symmetry?

We thank the reviewer for raising this important point. Crystals of Sema1a diffracted to a nominal resolution of 3.6 Å, and the diffraction data displayed strong anisotropy in its diffraction limits. For anisotropy correction, we used the Staraniso server which yielded an ellipsoidal resolution boundary with limits of 4.2, 4.2 and 3.2 Å along the a*, b*, and c* axes, respectively. Thus, the anisotropy correction yielded a higher resolution (3.2 Å) along the c* axis. The completeness of the anisotropic data was naturally reduced by the ellipsoidal truncation, however, this did not affect model building and refinement. We agree with the reviewer that the change of resolution after anisotropic correction is not apparent in our crystallographic table; however, it is not entirely clear how to present resolution limits along three axes after anisotropy correction in the single field of the table as formatted for Nature Communications. Therefore, to clarify the anisotropy correction, we have added a footnote to the Supplementary Table 1 (in which we give the resolution limits along the three axes) and the following sentence to the Methods – Page17:

"Anisotropy correction of Sema1a₁₋₂ yielded an ellipsoidal resolution boundary with limits of 4.2, 4.2 and 3.2 Å along the a*, b*, and c* axes, respectively, while anisotropy correction of Sema1b₁₋₂ yielded an ellipsoidal resolution boundary with limits of 2.7, 2.7 and 3.5 Å along the a*, b*, and c* axes, respectively."

We apologise for not more clearly explaining our treatment of twinning in the original methods section. The reviewer is correct, Xtrige analysis revealed 3 merohedral twin operators:

1. -h,-k,l (twin fraction 0.277)
2. h,-h-k,-l (twin fraction 0.107)
3. -k,-h-l (twin fraction 0.099)

First, we performed twin refinement in the Phenix package with individual twin operators. Application of the first operator reduced R_{free}/R_{work} by 7.1/6.2%, application of the second operator reduced R_{free}/R_{work} by 2.3/2.9% and application of the third operator reduced R_{free}/R_{work} by 4.0/3.2%. We were not able to use multiple twin operators in the Phenix refinement because currently, Phenix supports only a single twin operator. We then performed twin refinement using multiple twin operators in Refmac; however, the overall refinement statistic resulting from this procedure in Refmac was not as good as when using the first single operator only in Phenix. The electron density maps resulting from the two procedures were essentially identical. Thus, for simplicity, we report the results using the first operator for twin refinement of Sema1a in Phenix. We have added text to the methods section (page 17-18) to clarify this point.

To confirm the space group, we performed the molecular replacement in Molrep using all space groups suggested by twinning and symmetry analyses in Xtriage. Indeed, the best score and contrast value was calculated exclusively for the P32 space group. Furthermore, no significant pseudotranslation was detected in Xtriage analysis.

3. It is not entirely clear what the MD simulations contribute to this study. The claim that "there is no significant difference in the dynamic behavior between monomeric Sema1b and dimeric Sema2a" is surprising, since the disulphide bond that dimerizes Sema2a will allow substantial conformational reorientations of the individual Sema2a molecules. I am not sure it is worthwhile to include these simulations in this study.

We agree with the reviewer that the results of our MD simulations do not reveal significant differences in the dynamic behaviour of the semaphorin structures. Crystal structures determined by us and others suggest that semaphorins are compact molecules; however, they have never been analysed by MD simulations. Our manuscript provides the first exploration of the conformational dynamics of semaphorins and demonstrates their rigidity. Furthermore, our discovery of monomeric Sema1b raised questions about the flexibility of the loop regions, which in dimeric semaphorins are involved in homodimerization. We hypothesized that higher flexibility of those loops might contribute an entropic penalty that prevents Sema1b dimerization. However, this hypothesis was not confirmed by the simulation, and thus, we wrote that there is no significant difference in the dynamic behaviour between monomeric Sema1b and dimeric Sema2a.

We have now clarified this point in Results – page 6.

Reviewer #3:

The manuscript by Jones and colleagues is a highly interesting and important paper. In particular, Semaphorins (Semas) are the largest family of guidance cues (with over 20 family members conserved from invertebrates to vertebrates), playing multiple different critical roles in most, if not all tissues. Yet, we still have a poor understanding of how Semaphorins work, the interaction/integration of different semaphorins, and how the different Semaphorins work to activate their Plexin receptors. Here, the authors using the important Drosophila model, determined crystal structures for all members of Drosophila class 1 and 2 Semaphorins (i.e., 4 out of the 5 Semaphorins present in Drosophila). This strategy was very fruitful and among a range of important observations on the Semaphorins that emerged from the structure - their work revealed that Semas use a disulphide bond to stabilize dimer formation of Sema domains. The authors structural and biophysical data also suggest that one

of these Semaphorins, Sema1b, is a monomer, indicating that semaphorin function appears not to be restricted to dimers. The authors also, for the first time, demonstrate that semaphorins are able to form heterodimers with members of the same semaphorin class. These are also very interesting results since heterodimerization provides a potential further mechanism to allow for fine-tuning of Semaphorin function and cell-cell signaling.

In total, therefore, this is significant work and will be of high-value to the multidisciplinary community. Further, the authors have done a nice job of assembling the work - nicely combining structure, biophysical approaches, and biochemistry. The writing and data presentation is also clear with a lot of great extras and insights - including supporting heterodimerization of Semas for the first time. And thus, while there is no functional/physiological testing of their structural/biochemical findings, the manuscript stands on its own (with all the data from the 4 structures) - and thus I believe functional/physiological studies go beyond the scope of the present study and should await future analysis.

However, with that being said, I believe a few additional experiments would complete and add to the value of the manuscript (and the authors have the reagents already in hand to address these questions):

1) Related to Fig. 1F, was binding seen between Class 1 Semas and PlexB or Class 2 Semas and PlexA? This would be important to test. The authors have the protein and the data should be presented in Suppl Figures. Especially since the authors say "Drosophila class 1 semaphorins have been reported to bind PlexA while class 2 semaphorins has been shown to bind PlexB11-13. Therefore plexin binding is semaphorin class-specific, but within class seems to be relatively promiscuous." This is also very important based on Fig. 1G and the authors statement "Notably, the potential molecular determinants of semaphorin class specificity for PlexA or PlexB include substitutions of hydrophobic residues with polar residues and vice versa at positions 120, 180, 223, 218, and 243 (considering Sema2a numbering) and further substitutions of charged versus uncharged residues at positions 217, 244, 251 and 252 (Fig. S1)." And also that "Conversely, there is a high degree of sequence conservation within classes at the putative binding site, in particular for the class 2 semaphorins."

We thank the reviewer for pointing out that we are in a position to test for cross-class ligand-receptor promiscuity. To test for cross-reactivity between class 1 semaphorins and PlexB or class 2 semaphorins and PlexA, we produced the same panel of proteins described in our original manuscript and performed the MST binding experiments. As expected from the literature on semaphorin-plexin biology, we did not observe any measurable indication of PlexA or PlexB binding to class 2 or class 1 semaphorins, respectively. These MST binding experiments are shown in Supplementary Fig 6 and we note these results on page 7-8.

2) The authors demonstrate that semaphorins can form heterodimers with members of the same semaphorin class. What about between classes (such as the ecto domain of Sema-1a and Sema-2a or Sema-2b)? For similar reasons as noted above for 1), these two different possibilities should be tested (and also whether Sema-1aEcto and Sema-1bEcto can dimerize), and the authors have the tools in hand.

We again thank the reviewer for this useful suggestion. We have extended our analyses to test for semaphorin heterodimerization between Sema1a and Sema2a; however, we were not able to detect the heterodimer. The following paragraph has been added to the Results-Page12:

"We further investigated semaphorin heterodimerization between different semaphorin classes. We selected Sema1a and Sema2a as candidates for potential interclass heterodimerization as both Sema1a and Sema2a are disulphide linked dimers that are formed by an intermolecular disulphide bridge located at the same position in the sema domain. To examine the ability of Sema1a and Sema2a to heterodimerize, we transiently co-transfected HEK293T cells with Sema1a_{ecto} and Sema2a_{full} constructs encoding His6 and TwinStrep tags, respectively. Using two-step affinity chromatography, we pulled down Sema1a_{ecto} and Sema2a_{full} homodimers, however, we were not able to detect the Sema1a/2a heterodimer (Supplementary Fig. 8) supporting our hypothesis that the heterodimerization between different semaphorin classes appears to be unlikely."

We have not tested the ability to heterodimerize between Sema1a_{ecto} and Sema1b_{ecto} as our analysis of heterodimerization is based on matching disulphide-linked dimer architectures.

3) The authors say "Despite the low propensity to non-covalently dimerize, Sema1a_{ecto} maintains the homodimeric architecture by reason of the covalent interchain disulphide bond." Yet, it looks like the class 6 semas (which would be vertebrate orthologues of class 1 Semas) do not have the Cys for dimerization (Fig. S4). How do they dimerize? There is a little discussion of this but this should be expanded in the discussion. Also, on a related note, the authors should comment on whether some caution related to the strong conclusions (in the abstract, intro, discussion, and Fig. 6) on Sema1b and its working as a monomer might be warranted. For example, since Class 1 Semas are transmembrane (and are likely to function as transmembrane ligands in vivo), might there be roles for the transmembrane or intracellular regions of Sema-1b in dimerization (i.e., can the authors exclude that)?

In response to the reviewer's suggestions we have modified text throughout the manuscript to make clear that our data reveal that the Sema1b extracellular region (specifically the sema domain) has no detectable propensity to dimerize. This characteristic is clearly at the opposite extreme to the covalently linked sema domains of the dimeric Sema1a, Sema2a and Sema2b proteins. We believe this is an important observation, however, the reviewer is correct to advise caution as to how we word our discussion of it. We have added and modified text on page 14 to take due account of this advice. The class 6 semaphorin Sema6A sema domain does show detectable dimerization in solution and crystallises as a dimer using the canonical dimerization interface, however, previously reported biophysical measurements indicate that it can sample monomer and dimer states. We have expanded and clarified our discussion of these properties on page 15. We have also introduced further caution when bringing together our comparison of class 1 and class 6 semaphorins in the statement (page 15) "It is therefore intriguing that whilst *Drosophila* class 1 semaphorins may occur on the cell surface as a predominantly monomeric Sema1b and disulphide-linked dimeric Sema1a, their mammalian Sema6 homologues can potentially form an equilibrium of monomers and non-covalent dimers."

Minor:

1) The title for the legend of Fig 1 states "Crystal structure of *Drosophila* Semaphorins". However, since one of the *Drosophila* Semaphorins, Sema5c, is not included, the authors should change the title to "Crystal structures of Class 1 and Class 2 Semaphorins" to not be misleading.

This is a good point, we have changed the title for the legend of Fig1 to "Crystal structures of class 1 and class 2 semaphorins".

2) Lines 142-144: "In class 1 semaphorins, two distinct loops with no obvious structural and

functional role are substantially extended from the bottom face of the β -propeller, β 5A- β 5B and β 7A- β 7B in Sema1a, and β 2C- β 2D and β 6C- β 6D Sema1b (Fig S3A)." This sentence refers to Fig S3B, not Fig S3A, and so should be corrected.

Thank you, we have corrected Fig S3A to Fig S3B.

3) Lines 153-155: "Intriguingly, an area of the surface, corresponding to a co-receptor (neuropilin) binding site in the vertebrate Sema3s (ref 27), is distinguished by strong negative charge in the class 1 semaphorins (Fig. S5)." While this region is nicely mapped in yellow in the figure (Fig. S5), for the general reader, the authors should indicate here in the main text if and how this differs from the electrostatic potential of the neuropilin binding site in the vertebrate Sema3s. This is particularly interesting since neuropilin is not present in *Drosophila*, but a related protein could serve the same function.

We agree with the referee that this is an interesting observation and thank them for pointing out the need for clarification. We have revised the text to address this point (page 7).

4) Lines 162 -164 "In a microscale thermophoresis binding experiment, Sema2bfull showed the tightest binding among all *Drosophila* semaphorins, interacting with PlexB1-4 with an apparent K_d of 3 nM 164 (Fig. 1G)." This sentence refers to Fig 1F, not Fig 1G, and so should be corrected.

Thank you, we have corrected Fig 1G to Fig 1F.

5) A lower affinity was observed for class 1 semaphorins. Fc tagged dimerized Sema1aecto interacted directly with PlexA1-4 giving an apparent K_d of 141 nM while monomeric Sema1becto bound PlexA1-4 with a K_d of 586 nM." This sentence refers to Fig. 1F and so that should be included at the end of the sentence.

Thank you, we have added a call-out to Fig 1F at the end of the sentence.

REVIEWERS' COMMENTS:

Reviewer #2 (Remarks to the Author):

The authors have answered the issues I have raised, and I congratulate them with this nice piece of work.

Reviewer #3 (Remarks to the Author):

Thank you for your attention to my questions and further investigation. All looks good.